# The IgCAM CAR Regulates Gap Junction-Mediated Coupling on Embryonic Cardiomyocytes and Affects Their Beating Frequency

**DOI:** 10.3390/life13010014

**Published:** 2022-12-21

**Authors:** Claudia Matthaeus, René Jüttner, Michael Gotthardt, Fritz G. Rathjen

**Affiliations:** 1Max-Delbrück-Center for Molecular Medicine, Robert-Rössle-Str. 10, DE-13092 Berlin, Germany; 2Laboratory of Cellular Biophysics, NHLBI, NIH, 50 South Drive, Building 50 RM 3312, Bethesda, MD 20892, USA

**Keywords:** IgCAM, CAR, cardiomyocyte, cell–cell coupling, gap junction, beating frequency, embryonic heart

## Abstract

The IgCAM coxsackie–adenovirus receptor (CAR) is essential for embryonic heart development and electrical conduction in the mature heart. However, it is not well-understood how CAR exerts these effects at the cellular level. To address this question, we analyzed the spontaneous beating of cultured embryonic hearts and cardiomyocytes from wild type and CAR knockout (KO) embryos. Surprisingly, in the absence of the CAR, cultured cardiomyocytes showed increased frequencies of beating and calcium cycling. Increased beatings of heart organ cultures were also induced by the application of reagents that bind to the extracellular region of the CAR, such as the adenovirus fiber knob. However, the calcium cycling machinery, including calcium extrusion via SERCA2 and NCX, was not disrupted in CAR KO cells. In contrast, CAR KO cardiomyocytes displayed size increases but decreased in the total numbers of membrane-localized Cx43 clusters. This was accompanied by improved cell–cell coupling between CAR KO cells, as demonstrated by increased intercellular dye diffusion. Our data indicate that the CAR may modulate the localization and oligomerization of Cx43 at the plasma membrane, which could in turn influence electrical propagation between cardiomyocytes via gap junctions.

## 1. Introduction

The coxsackie–adenovirus receptor (CAR) is a 46 kDa cell adhesion protein of the Ig superfamily. It is composed of the membrane-distal V-type domain (D1) and the membrane-proximal C2-type domain (D2), which are connected by a short junction. The CAR shares sequence homology with the IgCAM CLMP (CAR-like membrane protein, also called ACAM, adipocyte adhesion molecule), ESAM (endothelial cell-selective adhesion molecule), and BT-IgSF (brain–testis IgSF, also called IgSF11 or VSIG-3) [1]. During embryonic development, the CAR is expressed in various organs, such as the brain, retina, liver, heart, pancreas, and lung, and is often found at cell–cell contact sites, including tight junctions [2,3,4,5,6,7,8]. Shortly after birth, the CAR expression significantly decreases in the nervous system and heart [2,5,9,10,11]. In the mature heart, the CAR specifically localizes to intercalated discs in close association with connexin 43 (Cx43), connexin 45 (Cx45), and zona occludens 1 (ZO-1) [12,13,14,15,16].

The absence of the CAR in mice causes malformation of the developing heart and hemorrhaging, leading to embryonic lethality around E11.5 to E13.5 [9,12,17]. Conditional inactivation of the CAR in the mature heart is associated with reduced expressions of Cx45 and Cx43, which disrupts electrical conduction between the atrium and the ventricle, and leads to atrioventricular block [13,16,18]. In addition, mice with CAR overexpression develop severe cardiomyopathy and die by 4 weeks of age [19]. Interestingly, the CAR is expressed at higher levels upon cardiac remodeling in patients suffering from dilated cardiomyopathy [20,21], during myocarditis [22,23], and after myocardial infarction [24]. The normally weak expression of the CAR in healthy adult cardiomyocytes, in contrast with its expression in the developing and diseased heart, implicates it in the formation of a functional myocardium and remodeling after injury [25]. The function of the CAR in the developing and adult nervous system, however, is still poorly understood. It is implicated in adult neurogenesis, synaptic homeostasis, synaptic plasticity, and/or neurite growth [2,26,27].

Adhesion and binding assays, as well as crystallographic studies, indicate that CAR promotes homophilic and heterophilic binding between neighboring cells [2,10,28,29]. The CAR exhibits heterophilic binding to extracellular matrix glycoproteins and other IgCAMs, such as JAML and JAM-C [2,30,31,32]. The cytoplasmic tail of the CAR contains a class I PDZ-binding motif for which several binding partners have been identified, such as ZO-1, MUPP-1 (multi-PDZ domain protein-1), MAGI-1b (membrane-associated guanylate kinase, WW, and PDZ domain containing 1b), PICK-1 (protein interacting with C kinase 1), the synaptic scaffolding protein PSD-95 (postsynaptic density protein 95), LNX (ligand-of-numb protein-X), and LNX2 [4,6,33,34,35,36].

Previous studies of the CAR have focused on its structural aspects and the physiological outcomes when the CAR is deleted in vivo. However, there is a lack of mechanistic insight into the involvement of the CAR in regulating the murine heartbeat. To better understand the molecular and cellular functions of the CAR, we analyzed its role in the spontaneous beating of embryonic murine hearts and monolayer cardiomyocyte cultures. We show that in the absence of the CAR, the beating frequencies of embryonic cardiomyocytes are increased. This was correlated with increased calcium cycling and calcium extrusion mechanisms. Further, the increased beating frequency was accompanied by increased gap junction activity, indicated by enhanced dye spread between CAR knockout (CAR KO) cardiomyocytes and increased sizes of Cx43 clusters. Taken together, our data indicate that the CAR is involved in the regulation of Cx43 and Cx45 localizations, which in turn affects the beating of cardiomyocytes at embryonic stages.

## 2. Materials and Methods

### 2.1. Mice

The mouse line CAR (B6.Cg-Cxadr^tm1^/Fgr) was described and genotyped, as detailed elsewhere [9]. Animals were housed on a 12/12 h light/dark cycle with free access to food. The animal procedures were performed according to the guidelines from directive 2010/63/EU of the European Parliament on the protection of animals used for scientific purposes. All experiments were approved by the local authorities of Berlin (LaGeSO) (numbers T0313/97 and X9014/15).

### 2.2. Embryonic Cardiomyocyte and Heart Organ Cultures

Global CAR-deficient mice die between embryonic days 11.5 and 13.5 [9]. Therefore, organ culture or dissociated cardiomyocytes were prepared from 10.5/11.5 day-old embryos of wild type or CAR mutant mice of the same litter. Intact hearts were cultured in 12-well chamber slides (Ibidi no. 81201). Hearts were seeded in 50 μL of Matrigel (Corning no. 356230) diluted 1:2 with DMEM/FCS (see also [37]) and incubated for 30 min at 37 °C in a humidified atmosphere of 95% air and 5% CO_2_. After solidification of the Matrigel, 350 μL of DMEM/FCS with or without the fiber knob (0.5 mg/mL) or the anti-CAR antibodies Rb80 (final concentration: 0.5 mg/mL of the IgG fraction) to mouse CAR or Rb54 to chick CAR were added. The IgG fractions of Rb80 or Rb54 were obtained by ProteinA (GE healthcare, Solingen, Germany) affinity chromatography. Antibodies were dialyzed against DMEM before application to cultures. Controls of the treated cultures contained vehicles and additional control cultures containing the IgG fractions of rabbit antibodies to the chick CAR (Rb54), which does not react with the mouse CAR [3]. The activity of the fiber knob Ad2 was controlled on CAR-deficient embryonic hearts.

Single cells were obtained by incubating embryonic hearts in 1 mg/mL of trypsin in 1 mM of EDTA/PBS (Invitrogen, Waltham, MA, USA) for 10 min at 37 °C followed by trituration in DMEM supplemented with 10% FCS and penicillin/streptomycin (Gibco). Dissociated cells from one embryonic heart were plated on one glass coverslip (diameter 12 mm) pre-coated with bovine fibronectin (10 μg/mL in PBS; Sigma-Aldrich, Darmstadt, Germany) in 50 μL DMEM/10% FCS/penicillin/streptomycin at 37 °C in a humidified atmosphere of 95% air and 5% CO_2_. When cells were attached after 4 to 5 h, 450 μL of culture medium was added and cells were cultured for 4 to 5 days before being measured. Cell or heart organ beatings were counted under an inverted microscope manually at 37 °C in a humidified atmosphere with 5% CO_2_. A total of 50% of the medium was exchanged every second day.

### 2.3. Preparation of the Fiber Knob of the Adenovirus

A cDNA encoding the fiber knob of the adenovirus Ad2C428N (abbreviated Ad2 in the following) was recombinantly expressed in bacteria and purified by sequential steps of ammonium sulfate precipitation, anion exchange chromatography, and Ni-NTA-Agarose (Qiagen no. 1018244) affinity chromatography, as described previously [2,38,39,40]. Before being applied to cultures, the fiber knob was dialyzed against DMEM.

For whole-mount staining of embryonic hearts, Ad2 was directly labeled via an NHS ester by the fluorescent dye Cy2 according to the instructions of the manufacturer (GE Healthcare, no. 32000). Ad2-Cy2 was applied at a concentration of 6 μg/mL in PBS/0.1%Tx100/5% goat serum for 2 h at room temperature to paraformaldehyde-fixed E10.5 hearts followed by several washing steps. Hearts were mounted in Immu-Mount (Thermo Scientific, Waltham, MA, USA, no. 9990402) for confocal imaging.

The binding of Ad2 to the fusion proteins CAR-Fc, CLMP-Fc, BT-IgSF-Fc, or Fc-fragment was done by the ELISA method [41]. A total of 200 μL of 1 μg/mL of the fusion were immobilized on ELISA plates (Nunc) and after blocking of residual binding sites on the plates, 200 μL of Ad2 at a concentration of 125 ng/mL was applied. The binding of Ad2 was detected by an HRP-conjugated monoclonal antibody to His-tag (Qiagen, Hilden, Germany) expressed by the fiber knob. O-Phenylenediamine (Sigma-Aldrich, Darmstadt, Germany) was used as a chromogen.

### 2.4. Calcium Imaging of Cultured Cardiomyocytes

Calcium transients of cultured embryonic cardiomyocytes at DIV4-5 were measured after incubating myocytes with the calcium-sensitive fluorescence dye Fura2-AM (1 μM, Biotium, Fremont, CA, USA, Cat: 50033-1) diluted in 0.02% Pluronic acid F-127/DMSO (Invitrogen, Waltham, MA, USA, P3000MP) in culture medium. After incubation for 60 min at 37 °C, cells were washed with DMEM, and calcium imaging analysis was performed in ACSF (artificial cerebrospinal fluid) (130 mM NaCl, 4 mM KCl, 1.25 mM NaH_2_PO_4_, 25 mM NaHCO_3_, 10 mM Glucose, 2 mM CaCl_2_, 1 mM MgCl_2_). Recordings were obtained with Zeiss Axio Examiner A1 and TILL Photonics LA software (version 2.3.0.18, 2013) at room temperature. The ratio (R) of Fura2 fluorescence intensity (R = F_340nm_/F_380nm_) was recorded by excitation at two wavelengths (340 nm and 380 nm, emission wavelength 510 nm, 10–20 ms exposure time, 4 × 4 binning) and was recorded; the analysis was carried out with Origin software (Additive, version 7.03, 2002), IgorPro (WaveMetrics, 2014, https://www.wavemetrics.com), and Fiji (ImageJ, version v1.49 m, 2015, https://imagej.nih.gov). To analyze the rate constants for the different calcium extrusion components (SERCA2, NCX, PMCA, and mitochondria), the protocol of Voigt et al. [42] was applied and described in detail in the Appendix A. Blockers were diluted in DMSO and added to the ACSF buffer before recordings started. The exposure times of the excitation and cycle times depended on Fura-2-loaded cells and the aim of the experiment. For the kinetic analysis of calcium transients (frequency, amplitude, decay time constant), the exposure times of 340 nm and 380 nm excitation wavelength were reduced to 10 ms, the cycle time was 42 ms and the total experiment duration was 1 min (23 frames/s). In the analysis, in the presence of specific pharmacological blockers, the exposure times increased to 20 ms, and the cycle time increased to 150 ms to prevent photobleaching (6.6 frames/s). The total duration time of an experiment depended on the particularly applied blocker but normally varied between 5 and 30 minutes. Pharmacological blockers for SERCA2 (thapsigargin, cyclopiazonic (CPA)), RyR receptor (tetracaine), and gap junctions (carbenoxolone (CBX)) were from Sigma-Aldrich.

### 2.5. Calcium Concentration Estimation

Cytosolic calcium was calculated according to the calcium calibration protocol published by Doeller and Wittenberg [43]. Briefly, cultured E10.5 cardiomyocytes were treated first with a calcium-free solution and followed by a high calcium solution during imaging recordings. Calcium-free or calcium-high solutions contained either 15 mM EGTA (Merck) or 25 mM CaCl_2_, diluted in ACSF, and supplemented with 10 μM of ionomycin. The intracellular calcium concentration was calculated according to the protocol by [43] whereby the K*_D_* value of 225 nM for Fura2-calcium was applied.

### 2.6. Analysis of Cell–Cell Coupling by Lucifer Yellow

The culture medium of cardiomyocytes was replaced by ACSF and 1% Lucifer Yellow (Invitrogen, L12926) in 90 mM KCl, 3 mM NaCl, 5 mM EGTA, 5 mM HEPES, 5 mM Glucose, 0.5 mM CaCl_2_, and 4 mM MgCl_2_) was carefully injected into an individual cardiomyocyte by a glass pipette via the CellTram injector (Eppendorf, Hamburg, Germany). After 5 min of dye spreading to neighboring cardiomyocytes, this was visualized by Zeiss Axio Examiner A1 using a 40× objective. Images were taken using TILL Photonics LA software (version 2.3.0.18, 2013). To measure the area of dye spreading, the recorded images from the injected cardiomyocyte and the dye spreading after 5 min were analyzed by Fiji (ImageJ, version v1.49m, 2015, https://imagej.nih.gov) software.

### 2.7. Whole-Cell PATCH-Clamp Recordings of Cultured Cardiomyocytes

To analyze voltage-gated Na^+^-currents in isolated cardiomyocytes, whole-cell patch-clamp recordings were carried out between 3 to 4 days in vitro. Cardiomyocytes were visualized under phase contrast optics on an upright microscope (Axioskop, Zeiss) by using a 63×/0.95 water immersion objective. Recordings were performed using a patch-clamp amplifier (EPC-9, HEKA Elektronik). Recording pipettes were filled with an intracellular solution containing (in mM): 10 NaCl, 120 KCl, 5 EGTA, 10 HEPES, 1 MgCl_2_, 1 CaCl_2_, and pH 7.3, 270 mOsmol/kg. The pipette-to-bath resistance ranged from 3 to 4 MOhm. Series resistance compensation was applied as much as possible (50–70%). The effective series resistance was in the range of 10–20 MOhm and was tested throughout the whole experiment by using a short depolarizing pulse (10 mV, 20 ms). Recordings were accepted only if the series resistance was less than 20 MOhm. The bath solution contained (in mM): 130 NaCl, 4 KCl, 15 glucose, 10 HEPES, 1 CaCl_2_, and 1 MgCl_2_ (pH 7.3, 310 mOsmol/kg). Whole-cell input resistance (R_IN_) was estimated on the basis of passive current responses to moderate depolarizing voltage pulses of short durations (±10 mV for 20 ms). Whole-cell membrane capacitance (C_m_) was estimated by integration of the capacitive current transient and division by the respective stimulation voltage. Voltage-activated Na^+^-currents were elicited by a series of 200 ms depolarizing pulses applied from the holding potential of −90 mV, in 10 mV increments between −90 and +50 mV. Passive responses were subtracted by using a hyperpolarizing pulse of −20 mV. Signals were acquired at a rate of 10 kHz and analyzed offline using WinTida 5.0 (HEKA Electronik, Harvard Bioscience, Hollison, KY, USA).

### 2.8. Microarray Analysis of mRNAs in the Embryonic Heart

CAR +/+ and CAR −/− E10.5 embryos were dissected and the hearts were flash-frozen. RNA was isolated from 3 pooled hearts/samples (*n* = 5; total number of analyzed hearts = 15 for each genotype) by using a Qiagen Mini RNA isolation kit according to the manufacturer’s protocol, followed by Agilent Bioanalyzer quality control. The Affymetrix Mouse Gene 1.0 ST and WT PLUS KIT (Affymetrix no. 902464) and a GeneChip Hybridization Wash and Stain Kit (no. 900720) were used according to the manufacturers’ protocols. The microarray was performed in cooperation with the Microarray Facility from the Max-Delbrück-Center. Datasets can be found at the Gene Expression Omnibus (GEO) with accession number GSE138831.

### 2.9. qPCR to Quantify the Level of mRNA of Selected Genes

Isolation of total RNA was performed from freshly dissected E11 hearts according to the instructor’s protocol (RNeasy Mini Kit, Qiagen). cDNA was generated with SuperScript II Reverse Transcriptase (Invitrogen). For the analysis of the expression levels of different genes, real-time PCR was performed using SYBR Select Master Mix (Applied Biosystems, Waltham, MA, USA) as detailed by the supplier. The relative fold change of the CAR KO gene expression compared to wild type hearts was calculated by using the comparative real-time PCR method [44,45]. Actin was used as the reference gene. See Appendix A for primers.

### 2.10. Biochemical Methods

Protein concentrations were determined using the Bradford assay (Bio-Rad no. 500-0006). Quantification of total proteins of the connexins in Western blotting was done by solubilizing individual E10.5/11.5 hearts in the SDS-PAGE sample buffer and centrifuging (10 min, 21,000× *g*). The band intensities were calculated using the software Quantity One (Bio-Rad). 

### 2.11. Immunoprecipitation

Immunoprecipitations were done by using covalently labeled IgG fractions of rabbit antibodies to mouse CARs (rb80) to agarose beads by sodium cyanoborohydride and by following the instructions of the Pierce Direct IP Kit (Thermo Scientific, no. 26148). A HeLa cell line stably expressing connexin43 [46] grown in DMEM/10%FCS/P/S supplemented with 1 μg/mL of puromycin (Sigma no. P9620) was solubilized in 1% Chaps/TBS/5% glycerol at pH 7.4 supplemented with protease blockers. Insolubilized material was removed by centrifugation (100,000× *g*, 10 min). Precipitation was done from 7 (HeLa Cx43) mg of solubilized proteins and 20 μg of immobilized anti-CAR IgG.

### 2.12. Immunocytochemistry and Immunohistochemistry

E10.5 embryos were fixed in 4% PFA/PBS for 1.5 h followed by incubation in 15% sucrose (Merck)/PBS for 2 h and overnight incubation in 30% sucrose/PBS. A total of 16 μm-thick cryostat sections were incubated in 0.1% Triton X-100/PBS/0.1% BSA/1% heat-inactivated goat serum using antibodies listed in Appendix A. Monoclonal mouse antibodies and corresponding secondary antibodies were incubated on sections using the MOM kit (Vector Laboratories, BMK-2202). Cells were counterstained with the nuclear marker DAPI (1 μg/mL) (Sigma). Immuno-stained cryostat sections or cultured cells were analyzed with the LSM 700 confocal microscope (Zeiss, using objectives 10×, 40×, 63×, and 100×), LSM Manager Software (Zeiss), and Fiji/Image J (Version v1.49m, 2015).

To quantify the connexin43 cluster and for staining, cardiomyocyte cultures were fixed in 4% paraformaldehyde in PBS for 5 min followed by solubilization using 0.1% TX-100/0.1% BSA in PBS and washing with PBS/BSA. mAb anti-sarcomeric actinin and rabbit anti-connexin43 and DAPI were applied in PBS/0.1%TX100/5% goat serum overnight. Contacting plasma membranes from two neighboring cells that contained connexin43 spots were encircled and connexin43 spots were counted in confocal images using Fiji software, setting the threshold to the RenyiEntropy routine; clusters were counted, with a size between 0.05 and 1 μm^2^ (2–40 pixels) [47]. The number of spots was related to the encircled area and the average value per cell was calculated. In total, 57 wild type and 67 knockout cardiomyocytes were analyzed from 3 embryos for each genotype, and roughly 20 spots per cell–cell contacts were counted. Then, the average sizes of connexin43 spots of each cardiomyocyte were calculated. Co-localizations of the CARs with ZO-1 or connexin43 on the plasma membrane regions of cultured cardiomyocytes were analyzed from confocal images by using Fiji/Image J. Co-localizations of the CARs with ZO-1 or connexin43 on plasma membranes of cultured cardiomyocytes were analyzed from confocal images by using Fiji/Image J. Calculation of the Pearson correlation was determined by the Colo2-derived intensity-based correlation analysis. Costes threshold regression was applied and the Pearson coefficient (P) above the threshold was used.

For whole-mount staining of the CAR, the IgG fraction of rabbit 80 was applied at a concentration of 2 μg/mL in PBS/0.1%Tx100/5% goat serum for 2 h at room temperature to paraformaldehyde-fixed E10.5 hearts followed by several washing steps and labeling with goat anti-rabbit-Cy3 and DAPI. Hearts were mounted in Immu-Mount (Thermo Scientific, no. 9990402) for confocal imaging.

### 2.13. Statistical Analysis

For statistical analysis of the data, SigmaStat software 3.5 (Systat Software, 2006) and GraphPad Prism were used. The Kolmogorov–Smirnov test was carried out to determine whether data were normally distributed. If datasets were normally distributed, t-tests or paired t-tests were applied to measure the significance between the groups of data. The heart organ cultures were analyzed by two-way ANOVA. Other data were analyzed with the Mann–Whitney–Rank-Sum test. The outlier test was performed online using GraphPad QuickCalcs https://www.graphpad.com/quickcalcs/grubbs1. The decay time constants were calculated by a script written in the software IgorPro (WaveMetrics, 2014). Data were represented as means ± SEM. If not given in the figure legends, the following *p*-values were used to indicate significant differences between two groups: * *p* < 0.05, ** *p* < 0.01, *** *p* < 0.001.

## 3. Results

### 3.1. The Absence of the CAR Resulted in Increased Spontaneous Beating Frequencies of Cultured Embryonic Cardiomyocytes

The CAR ablation has been shown to disrupt electrical conduction between cardiomyocytes in the adult heart and was correlated with reduced expressions of Cx45 and Cx43 [13,16,48]. In one of these mutant mice, an increase in the maximal heart rate to 870 in comparison to 730 beats per minute in the wild type was measured [16]. To analyze the role of the CAR in the beating of the murine heart at the cellular level, we cultured embryonic cardiomyocytes at high densities on fibronectin-coated coverslips. This culture system allowed us to examine cellular calcium cycling, as well as the electrophysiological properties of the CAR KO cardiomyocytes. Global CAR-deficient mice die between embryonic days 11.5 and 13.5 due to the malformation of the embryonic heart [9,12,17]. Therefore, we prepared cardiomyocytes from E10.5 hearts from littermates of wild type (WT) and CAR-deficient mice. Two days after plating, cardiomyocytes of both genotypes began to beat spontaneously and exhibited synchrony after three days. As indicated by anti-sarcomeric α-actinin staining, the CAR was uniformly localized to the surface of cardiomyocytes (Figure 1A). After 4–5 days in vitro (DIV), we assessed the beating frequencies of the cardiomyocytes using manual counting (Figure 1B) and calcium imaging with the ratiometric calcium indicator Fura2 (Figure 1C,D).

Surprisingly, CAR-deficient cardiomyocytes showed increased beating frequencies (96 ± 7 bpm) compared to WT cardiomyocytes (59 ± 5 bpm, Figure 1B). In line with these measurements, CAR KO cardiomyocytes exhibited significantly increased spontaneous calcium cycling, which correlated with a faster decline of individual calcium transients (Figure 1C,D). The calcium transients for CAR KO cells consistently showed significantly shorter decay time constants compared to WT calcium transients (Figure 1E). In addition, we observed significantly decreased decay time constants in mutant cardiomyocytes after caffeine application, which triggers a complete release of calcium stored in the sarcoplasmic reticulum (Figure 1F,G). Caffeine-induced transients are long-lasting in comparison to spontaneous calcium transients. 

These results suggest that the increased beating rates of cultured CAR-deficient embryonic cardiomyocytes could be linked to alterations in calcium cycling, intracellular calcium levels, calcium extrusion from the cytosol, calcium release from internal stores, or ionic currents. Alternatively, the increased frequency of calcium transients could result from impaired cell–cell communication. To distinguish between these possibilities, we examined calcium cycling, as well as the electrophysiological properties and cell–cell coupling of WT and CAR-deficient cardiomyocytes.

#### 3.1.1. Mechanisms of Calcium Cycling Were Not Impaired in CAR-Deficient Cardiomyocytes

As changes in intracellular calcium levels can influence the beating frequencies of cardiomyocytes, we next estimated the cytosolic calcium concentrations in WT and CAR-deficient cells, as detailed elsewhere [43]. Both genotypes showed calcium concentrations of 120–130 nM and 350–450 nM during diastolic and systolic phases, respectively, indicating no significant differences in intracellular calcium levels (Figure 2A). In addition, the total calcium content stored in the sarcoplasmic reticulum, which was deduced from the amplitudes of calcium transients after the application of 10 mM of caffeine, was similar in both genotypes (Figure 2B) [43]. Further, we observed no differences in the calcium leakage from the sarcoplasmic reticulum when blocking ryanodine receptors with 1 mM of tetracaine (Figure 2C).

Finally, we investigated calcium extrusion mechanisms in cardiomyocytes. Systolic calcium extrusion is mainly carried out by SERCA2 (sarcoplasmic/endoplasmic reticulum calcium ATPase) and NCX (sodium–calcium exchanger), with minor contributions from PMCA (plasma membrane calcium ATPase) and mitochondria [49]. To investigate all four calcium extrusion components, we calculated their rate constants as previously described [42] (see Appendix A for detailed descriptions and calculations). The rate constant (*k*) is defined as the amount of calcium per second that is removed from the cytosol and can be calculated as the reciprocal of the decay time constant (τ). In both WT and mutant embryonic cardiomyocytes, calcium was primarily extruded by SERCA2 (Figure 2D) or NCX when SERCA2 was inhibited by caffeine (Figure 2E). CAR KO cardiomyocytes, which we previously revealed to beat at faster rates, showed increased rate constants for both SERCA2 (grey bar) and NCX (red bar). PMCA and mitochondria only played minor roles in calcium extrusion for both genotypes during spontaneous beatings. Further, the relative amount of calcium (shown as %) removed by SERCA2, NCX, PMCA, or mitochondria was very similar in both genotypes (Figure 2D,E). In the cardiomyocytes of both genotypes, about 90% of the cytosolic calcium was removed by SERCA2. CAR KO cardiomyocytes showed slight increases in NCX-mediated calcium removal compared to WT cells (5% vs. 7%). Additional physiological studies for NCX using different extracellular sodium concentrations did not reveal any differences between CAR-deficient and WT cardiomyocytes (see Appendix A).

Interaction with phospholamban, which itself is controlled by phosphorylation, with SERCA2 negatively regulates calcium removal from the cytosol and could, therefore, affect the cardiomyocyte beating frequency [50,51]. However, we observed no differences in the compositions of di-, tri-, and pentamers between WT and CAR KO embryonic hearts (Figure 2F), indicating no deregulation of SERCA2 by phospholamban. Further, the SERCA2 blockers thapsigargin and CPA only partially reduced beatings in CAR KO cardiomyocytes, which might be a reflection of the increased beating activities in these cells (Appendix A).

#### 3.1.2. Electrophysiological Properties Were Not Changed in CAR-Deficient Cardiomyocytes

To analyze of electrophysiological properties of WT and CAR KO cardiomyocytes, we performed whole-cell patch clamp recordings on cultured cardiomyocytes. We observed no differences in I_Na_ current (Figure 2J–L) between WT and CAR KO cells (WT: −338.2 ± 47.6 pA/pF, *n* = 23 and CAR KO: −384.4.3 ± 50.7 pA/pF, *n* = 47, *p* = 0.5173, mean ± SEM, unpaired *t*-test). Consistently, the mRNA levels of Nav1.5 (Scn5a) and other sodium channels were unaltered between WT and CAR knockout embryonic hearts. The same held true for potassium channel mRNA transcripts (see database entry GEO GSE138831 and Appendix A). Further, the input resistance (WT: 435.1 ± 55.5 MOhm, *n* = 23 and CAR KO: 502.7 ± 67.2 MOhm, *n* = 34, *p* = 0.4492, mean ± SEM) and cell capacitance, which are usually taken as measures of cell sizes, were similar in both genotypes (WT: 51.9 ± 10.2 pF, *n* = 23 and CAR KO: 36.5 ± 3.5 pF, *n* = 34, *p* = 0.175, mean ± SEM). These data indicate that the increased beating frequencies of CAR KO cardiomyocytes are likely not caused by changes in their electrophysiological properties. 

Taken together, we conclude that calcium cycling, the associated extrusion machinery, and the electrophysiological properties are not disrupted in the absence of the CAR. The increased activity in calcium signaling that we measured in CAR KO cardiomyocytes may instead be a reflection of the increased beating frequencies in these cells.

### 3.2. Gap Junction-Mediated Coupling Is Increased in CAR-Deficient Cardiomyocytes

The propagation of electrical activity between cardiomyocytes and, consequently, the beating of these cells, depends on the expression and regulation of gap junctions [54]. Therefore, we asked whether gap junction-mediated communication might be impaired in CAR-deficient cardiomyocytes. Cx43 and Cx45 are major connexins expressed in embryonic myocytes [55]. In both WT and CAR-deficient cardiomyocytes, Cx43 was primarily found at contact sites between neighboring cells (Figure 3A). Surprisingly, these Cx43 clusters were significantly larger in CAR KO cells (on average, 0.292 μm^2^ and 0.3352 μm^2^ for WT and knockout, respectively; Mann–Whitney U test (*p* = 0.0007)) (Figure 3B). The relative number of small clusters per membrane area was consistently reduced in CAR KO cardiomyocytes compared to WT cells (Figure 3C). On average, we observed a 40% decrease in total Cx43 clusters at cell–cell contact sites in CAR-deficient cardiomyocytes (Figure 3D). The increase in large Cx43 clusters in CAR KO cells was not accompanied by altered activity of the kinase Akt, which is known to increase the gap junction size by phosphorylating specific serine residues on Cx43 (Figure 3E) [56].

Given the increase in size but the decrease in the total number of Cx43 clusters in CAR KO cardiomyocytes, we next assessed cell–cell coupling using a dye diffusion assay. We microinjected WT and CAR KO cardiomyocytes with Lucifer Yellow, a fluorescent tracer that passes through gap junctions. After 5 min, the glass electrode was withdrawn, images were taken, and the number of recipient cells (Figure 3G) and spread area (Figure 3F) were measured to quantitatively determine gap junction communication. In comparison to WT cardiomyocytes, CAR KO cells showed a significantly increased number of coupled cells receiving dye from donor cells (Figure 3G), and an increase in the dye spread area (Figure 3F). Importantly, coupling was inhibited in all genotypes by the gap junction blocker CBX (200 μM, applied 10 min before the Lucifer Yellow application) (Figure 3H).

Further, we observed reductions in Cx43 and Cx45, but not Cx40, Cx50, or β-catenin, at the protein levels in CAR-deficient hearts by Western blotting (Figure 4A,B). The mRNA levels of Cx43 and Cx45, as well as the number of other genes, including cell–cell adhesion components and cytoskeletal elements, remained unchanged (Figure 4C,D and Appendix A, see also database entry GEO GSE138831). qRT-PCR experiments performed for a number of selected genes confirmed the microarray data (Appendix A), indicating that changes occurred at the protein and not at the level of mRNA.

Taken together, these observations indicated a higher degree of cell–cell coupling between CAR-deficient cardiomyocytes compared to WT cells. This suggests that improved electrical propagation via gap junctions could accelerate the beatings of CAR-deficient cardiomyocytes. These data are in line with observations that CAR-deficient adult hearts exhibit increased dye coupling between cardiomyocytes and a reduced level of Cx43 protein [13,16].

Since the CAR appears to affect the oligomerization status of Cx43 at the plasma membrane, we investigated a possible direct interaction between the CAR and Cx43. In cultured cardiomyocytes, the CAR was found in regions of the plasma membrane where Cx43 was not present. Quantification of the co-localization data using the Pearson correlation analysis suggested that the CAR might not interact directly with Cx43 (Figure 4E,F). Consistently, we found no co-immunoprecipitation of the CAR and Cx43 using a HeLa cell line that stably expresses Cx43, suggesting that the CAR exerts its effect on Cx43 indirectly (Figure 4G). For comparison, the localization of the scaffolding proteins ZO-1 and CAR, which are known to bind to each other [4], are shown (Figure 4E,F).

### 3.3. Increased Beating of CAR-Deficient Embryonic Hearts in Organ Cultures

In the embryonic heart, the CAR is expressed on the surfaces of all cells (Figure 5A,B), but at postnatal stages, becomes predominantly restricted to the intercalated discs (Figure 5B, depicted by arrowheads) (for further postnatal stages, see [25]). To extend our observations of the increased beating activity of CAR-deficient cardiomyocytes in monolayers, we cultured intact E10.5 hearts from the littermates of WT or CAR-deficient mice in Matrigel on chamber slides for up to 72 h, and the frequency of beating was determined at different time points (Figure 5C). In addition, we applied polyclonal antibodies to the extracellular domain of the CAR or “fiber knob” to explant cultures. Both reagents are known to disrupt cell–cell contacts in neurons [3]. “Fiber knob”, here referred to as Ad2, is the tip of the homotrimeric protein of the fiber from the adenovirus capsid, which binds the CAR on the host cell surface for the infection (Figure 5D,E,G). Ad2 binds up to three D1 polypeptides of the CAR [57,58] with an affinity higher than CAR to itself and interferes with the cell–cell contact formations of epithelial cells and neurons [3,59]. It interacts specifically with the CAR but not with the highly-related proteins CLMP and BT-IgSF (Appendix A) [60].

The regular beating of explanted hearts started after four to five hours in vitro. We detected a significant increase in the average spontaneous beating frequency of explanted hearts lacking the CAR over several days in culture (Figure 5F). Similarly, the application of Ad2 or polyclonal antibodies to the CAR also resulted in an increase in the average spontaneous beating frequency in WT hearts (Figure 5G,H). This suggests that the disruption of CAR’s homophilic binding activity, induced by the addition of Ad2 or CAR antibodies, could induce the same effects seen in the CAR KO. Specificity in this culture system was demonstrated by applying antibodies to the chick CAR (Rb54), which does not bind to the mouse CAR [3] (Appendix A). Taken together, we conclude that the CAR may act as a regulator in gap junctions and heart beating.

## 4. Discussion

The goal of the present study was to examine the function of the Ig cell adhesion molecule CAR in embryonic cardiomyocytes. Unexpectedly, we observed that CAR KO embryonic heart organs and cardiomyocyte cultures exhibited increased spontaneous beating frequencies compared to their WT counterparts. This was demonstrated by manual counting and an analysis of calcium cycling, including the mechanisms of calcium extrusion via SERCA2 and NCX. In CAR KO cardiomyocytes, the SERCA2 and NCX rate constants, as well as the proportion of cytosolic calcium removed by SERCA2 and NCX, were significantly increased. However, we observed no changes in the calcium cycling machinery or the electrical properties of these cells. In contrast, the increased beating correlated with increased gap junction dye coupling and increased sizes of Cx43 clusters on CAR-deficient cardiomyocytes.

The propagation of electrical activity between cardiomyocytes, which in turn influences the beating of these cells, depends on the regulation of gap junctions [54]. The gating of gap junctions containing Cx43 or Cx45 is primarily regulated by factors such as voltage, pH, and phosphorylation [61]. Cell–cell coupling, which is itself modulated by the size and localization of gap junctions, is also crucial for coordinating the spread of action potentials and calcium waves in the heart [62]. Increased cardiomyocyte–cardiomyocyte coupling is driven by an increase in the junction size could, therefore, impact the beating frequency [63]. Although the amounts of connexins 43 and 45 proteins were decreased in the absence of the CAR, the remaining Cx43 formed larger clusters. Based on the increased sizes of Cx43 clusters in CAR KO cardiomyocytes, we hypothesize that electrical excitation pulses in CAR-deficient cardiomyocytes might more rapidly transfer to neighboring cells, which in turn might result in their faster depolarization. Importantly, our data corroborate the results of a previous study in which CAR-deficient adult hearts showed increased cardiomyocyte coupling and a decrease in Cx43 and Cx45 levels [13,16]. Here, the maximal heart rate raised from 730 beats per minute in the wild type to 870 beats per minute (120%) in the mutant mouse [16]. A role for connexins in regulating cardiomyocyte beating frequency was also shown for Cx45 KO embryonic stem cell-derived cardiomyocytes [64] and in a model with conditional deletion of Cx43 in the mature heart. This resulted in a reduction of the conduction velocity [65,66]. Dysfunction and malformation of the developing heart have also been observed in the absence of Cx43, Cx45, and Cx40, which disrupted the gap junction communication [67,68,69,70,71].

The effects of the CAR KO shown here are reminiscent of those observed in knockouts of the CAR-related cell adhesion proteins CLMP and BT-IgSF. In mature CLMP-deficient mice, Cx43 and Cx45 levels are reduced in smooth muscle cells of the intestine and ureter, resulting in uncoordinated contractions [47]. In BT-IgSF knockouts, Cx43 is mislocalized in the Sertoli cells of the testes and reduced in astrocytes, which impairs cell–cell coupling between astrocytes in the hippocampus or cortex slices [72,73]. Together with our observations that the CAR plays also a cell–cell coupling role in the adult heart, our present data suggest that the CAR and the highly-related cell adhesion proteins CLMP and BT-IgSF might regulate the localization and oligomerization status of Cx43 [1].

An interesting topic for future research involves the mechanism by which CAR exerts its influence on Cx43. Since connexin mRNA levels were not reduced in the CAR KO, the CAR might provide a signal that facilitates Cx43 biosynthesis or might stabilize the expression or oligomerization status of Cx43 within the plasma membrane. However, a direct interaction is unlikely as we observed no co-localization or co-immunoprecipitation of the CAR with Cx43. The scaffolding protein ZO-1 is also known to regulate the gap junction size [74]. ZO-1 directly interacts with the C-terminal segment of Cx43, which might include its second PDZ domain [75,76,77]. Inhibition of this binding leads to the uncontrolled formation of large gap junction clusters in the culture [74,78,79]. We observed co-localization between CAR and ZO-1 in cultured embryonic cardiomyocytes, which has previously been demonstrated in epithelial cells [4,6]. Although we could not co-precipitate CAR and ZO-1, which might by due to the harsh extraction conditions, it is still conceivable that CAR could influence the Cx43 gap junction size, in part, via ZO-1. For example, the absence of the CAR might cause inhibition of the binding between Cx43 and ZO-1, which in turn might promote the uncontrolled growth of gap junction clusters.

Interestingly, we observed that reagents that bound to the extracellular region of the CAR, such as Ad2, interfered with the beating of embryonic hearts in the culture. Crystal structures have been solved for the complete CAR extracellular domain, as well as a complex consisting of the CAR N-terminal-located V-type domain of the CAR bound to Ad2 [2,28,29,57]. Conserved amino acid residues within the GFCC′C″ surface (650 Å^2^) of the CAR V-type domain are implicated in the homophilic binding of the CAR. This area of the CAR overlaps with the region that interacts with Ad2 [57]. The stimulatory effect of Ad2 on the beating of heart organ cultures might, therefore, result from the disruption of CAR–CAR interactions between neighboring cells. Further studies will be required to determine if trans-homophilic binding of the CAR can modulate the organization of connexins in cardiomyocytes. In summary, we conclude that CAR may regulate gap junctions in murine cardiomyocytes. Candidates that might target CAR in this context include full-length (or fragments of) AD2, and nanobodies specific to the CAR extracellular domain.

## Figures and Tables

**Figure 1 life-13-00014-f001:**
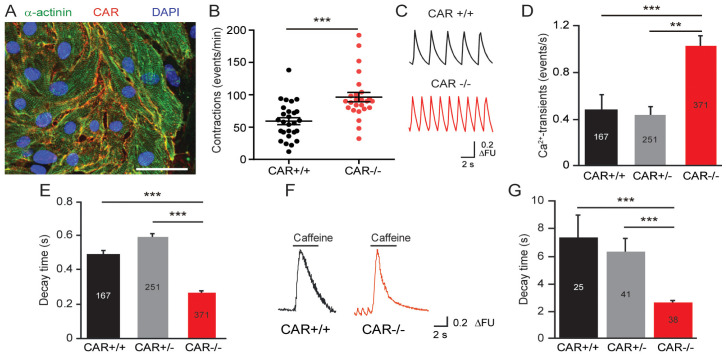
Increased frequency of beating and calcium transients in monolayer cultures of embryonic cardiomyocytes in the absence of the CAR. (**A**) Localization of the CAR in cultures from embryonic E10.5 hearts cultured 4 DIV in DMEM/FCS. Cardiomyocytes were identified by labeling with an anti-sarcomeric actinin antibody (green). Scale bar, 80 μm. (**B**) The beating frequencies of cardiomyocytes were counted manually at DIV 4 at 37 °C in a humidified atmosphere of 95% air and 5% CO_2_ in DMEM/FCS by using an inverted microscope. Cells were from 13 independent cultures and 27 wild type and 25 knockout cell clusters were counted. (**C**) Individual traces of calcium transients of cultured cardiomyocytes are shown. FU, relative fluorescence units. (**D**) Summary of the frequency of calcium transients from wild type, heterozygote, and CAR-deficient cardiomyocytes. Measurements were done from Fura-2-loaded cardiomyocytes in ACSF at room temperature. The numbers in the columns represent the number of cell clusters from independent cultures. (**E**) Decay time constants of calcium transients of wild type, heterozygote, and CAR-deficient cardiomyocytes. (**F**,**G**) Individual traces of caffeine-induced calcium transients and their decay time constants are shown. (** *p* < 0.01, *** *p* < 0.001).

**Figure 2 life-13-00014-f002:**
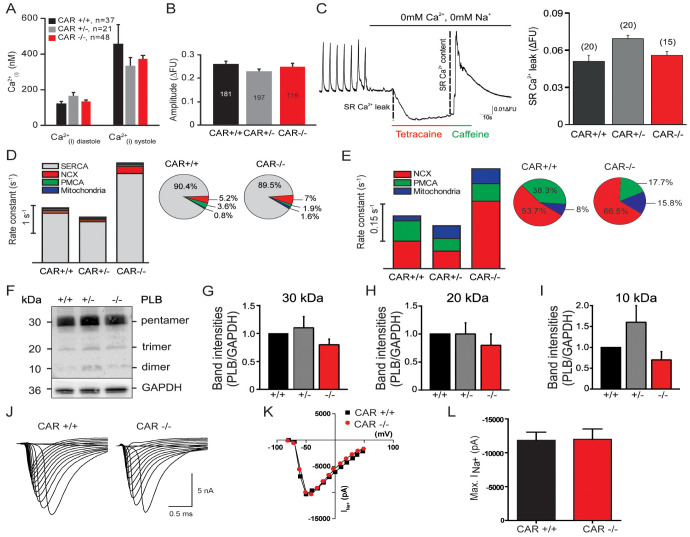
Calcium extrusion is enhanced in CAR knockout cardiomyocytes, which correlates with the increased beating frequency. Calcium extrusion is enhanced in CAR knockout cardiomyocytes, which correlates with the increased beating frequency. (**A**) Intracellular calcium concentrations at diastolic and systolic phases of wild type, heterozygote, and CAR-deficient cultured cardiomyocytes are depicted. (**B**) Amplitudes of calcium transients after the application of 10 mM of caffeine, which might be considered a measure of calcium levels stored in the sarcoplasmic reticulum. No differences were observed between the different genotypes, suggesting that similar amounts of calcium are stored in the sarcoplasmic reticulum. (**C**) Spontaneous calcium events via the RyR, so-called calcium leaks, can induce spontaneous beating and fibrillation of the heart [42,52,53]. The Ca^2+^ leak can be quantified by measuring the changes of cytosolic Ca^2+^ in the presence of 1 mM of tetracaine (RyR inhibitor) in Na^+^- and Ca^2+^- free ACSF. Tetracaine completely blocked the RyR and, therefore, the SR Ca^2+^ leak, consequently, the cytosolic Ca^2+^ concentration decreased. Wild type and CAR knockout cardiomyocytes did not show any significant difference in the intracellular Ca^2+^ drop. Further control at the end of the experimental caffeine was applied to the cells, which induced a complete SR release of Ca^2+^. There was no difference in the Ca^2+^ transient amplitude between wild type and CAR knockout cardiomyocytes. (**D**) Rate constants for all four calcium extrusion components were calculated for spontaneous calcium transients (for calculations, see Appendix A). SERCA2 and NCX showed significantly increased rate constants in CAR knockout cardiomyocytes. The relative amounts of calcium removed by the different mechanisms are shown for both genotypes in percentages at the right. (**E**) In caffeine-induced calcium transients, the NCX rate constants were significantly increased in CAR knockout cardiomyocytes compared to CAR wild types. The relative amounts of calcium removed by the different mechanisms are shown for both genotypes in the percentages. (**F**–**I**) Western blot analysis of the band intensities of phospholamban (PLB) did not show any differences between CAR wild type and knockout E10.5 hearts. Quantification of band intensities of the pentamer (**G**), trimer (**H**), and dimer (**I**) were calculated. Since the regulation of SERCA2 by phospholamban might affect the beating frequency, we analyzed the polypeptide composition of phospholamban in the absence of the CAR. The interaction of phospholamban with SERCA2 negatively regulates calcium removal from the cytosol. Phosphorylation of phospholamban by PKA results in a release of the monomers from SERCA2, which then reassemble into di-, tri-, or pentamers [50,51]. (**J**–**L**) Sodium currents are not altered in CAR-deficient cardiomyocytes. Single traces (**J**), the current/voltage relationships (**K**), and the summary (**L**) of whole-cell patch clamp recordings of cultured cardiomyocytes are depicted.

**Figure 3 life-13-00014-f003:**
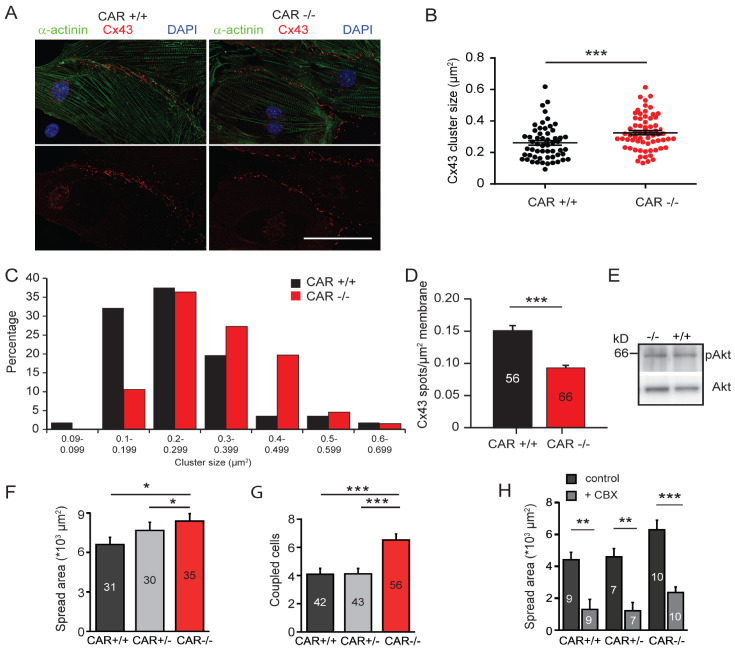
The sizes of connexin43 spots and dye coupling are increased in the absence of the CAR. (**A**–**D**) Quantification of connexin43 spots at cell–cell contact sites of cultured E10.5 wild type and CAR-deficient cardiomyocytes. Cells were stained by rabbit anti-connexin43, mAb anti-sarcomeric actinin, and DAPI. The average sizes of connexin43 clusters increased; (**B**) data are from three independent cultures, Mann–Whitney test *p* = 0.0007. In total, 56 wild type and 66 knockout cardiomyocytes were analyzed. Roughly 20 spots per cell were counted and the average size per cell was calculated. (**C**) The size distribution of connexin43 plaques shows a shift toward a larger cluster while smaller ones are reduced (chi-square test *p* = 0.0182). (**D**) The total number of connexin43 plaques per cell–cell contact area was reduced. (Mann–Whitney U test *p* < 0.001). Scale bar in A, 50 μm. (**E**) Western blot of wild type and CAR-deficient embryonic hearts using antibodies to the phosphorylated Akt (pAkt) or total Akt (lower panel) are shown. (**F**,**G**) Individual cardiomyocytes from the E10.5 CAR wild type and CAR knockout cultures were injected with Lucifer Yellow and the spreading was examined after 5 min. The spread area (**F**) and the number of coupled (Lucifer Yellow stained) cardiomyocytes (**G**) were significantly increased in CAR knockout cultures. (**H**) Lucifer Yellow spread was blocked by the gap junction blocker carbenoxolone (CBX) at 200 μM. (* *p* < 0.05, ** *p* < 0.01, *** *p* < 0.001).

**Figure 4 life-13-00014-f004:**
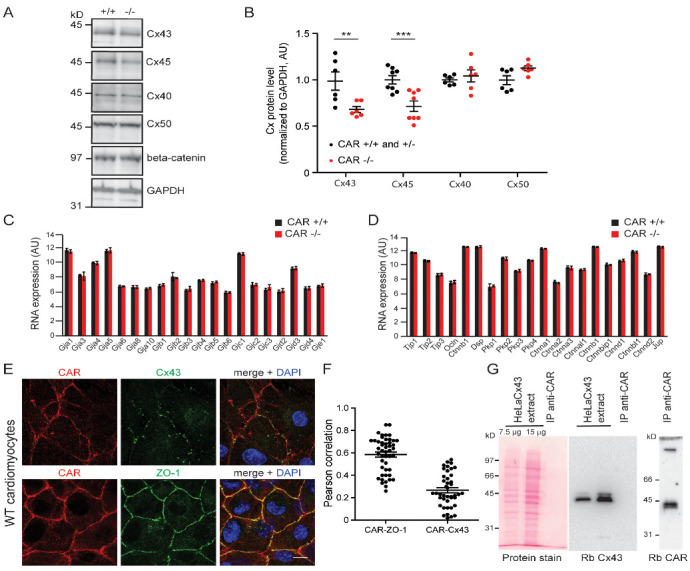
Decreased expressions of connexin43 and 45 in CAR-deficient embryonic hearts. (**A**,**B**) Expressions of connexin43, 45, 40, and 50 in wild type and CAR-deficient E10.5/E11.5 hearts shown by Western blots. For comparison, β-catenin and GAPDH are shown (from top to bottom). Band intensities from blots were quantified. Reductions were observed for connexin43 and 45 (*t*-test *p* = 0.0095 and *p* < 0.0001, respectively) but not for connexins 40 or 50 (*t*-test 0.5773 and 0.1179, respectively) in knockouts. Individual hearts were directly solubilized in the SDS-PAGE sample buffer. Molecular mass markers are indicated on the left of the panel. AU, arbitrary units. (**C**,**D**) Selected Affymetrix microarray data of mRNA expression levels of connexins and cell–cell contact proteins of E10.5 hearts of wild type and CAR-deficient mice did not reveal differences. For further details, see Appendix A and the database entry (GEO GSE138831). (**E**,**F**) On E10.5 cardiomyocytes, the CAR was not found to co-localize with connexin43. ZO-1 is shown for comparison. Scale bar, 10 μm. The Pearson correlation of co-localization was calculated using Fiji/Image J software (**F**); 48 and 43 images were analyzed for CAR-ZO-1 and CAR-Cx43 co-localization, respectively. (**G**) No co-immunoprecipitations of connexin43 and CAR were detected. (** *p* < 0.01, *** *p* < 0.001). The CAR was precipitated from extracts of HeLaCx43 by the IgG fraction of rabbit 80 (anti-CAR) directly coupled to CNBr-activated beads. The left panel shows the detergent extracts of HeLaCx43 cells stained by Ponceau. (Please note that this protein stain is not sensitive enough to detect proteins obtained by immunoprecipitation). The middle pellet shows the blot with anti-Cx43 of the HeLaCx43 extract and the IP with anti-CAR. No connexin43 could be detected; however, the CAR was easily visualized in the IP (right panel). A total of 7 mg of total protein of the HeLaCx43 extract and 20 μg IgG were used in the IP.

**Figure 5 life-13-00014-f005:**
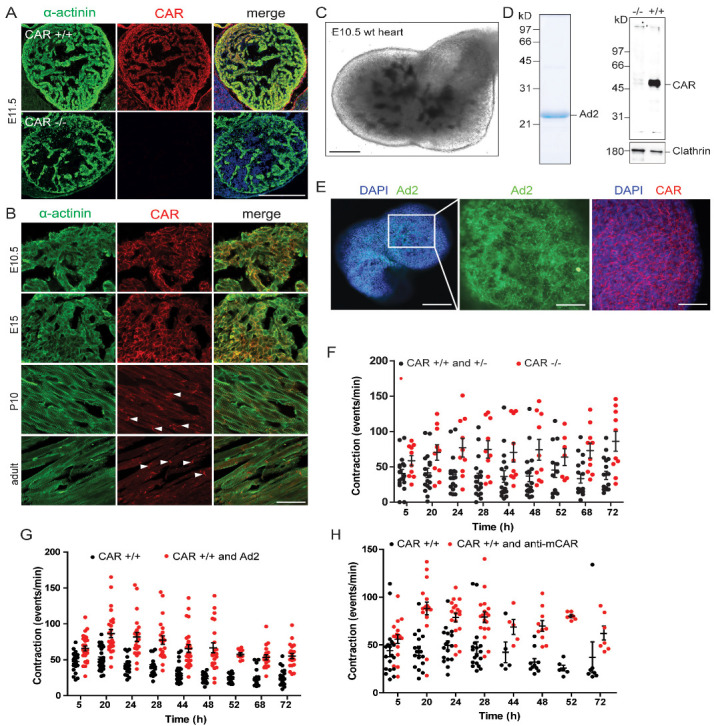
Localization of the CAR in the developing and mature heart and the increased spontaneous beating frequency of heart organ cultures in the absence of the CAR or the presence of the fiber knob or antibodies to CAR. (**A**) Overview of the localization of the CAR in the heart at E11.5. Cryostat sections were stained by rabbit anti-mCAR generated against the extracellular domain of the CAR (Rb80) and by a monoclonal antibody against sarcomeric α-actinin. Sections from knockout hearts indicated the specificity of Rb80. Scale bar, 100 μm. (**B**) Localization of the CAR at different developmental stages and in the mature heart. Arrowheads in sections of P10 or the adult heart indicate intercalated discs. Scale bar, 50 μm. (**C**) In vitro culture of E10.5 heart in Matrigel to determine the beating rate. Scale bar, 200 μm. (**D**) Purified recombinantly expressed fiber knob Ad2 is shown in 12% SDS-PAGE stained by Coomassie blue. Specificity of the rabbit antibody to the extracellular region of mCAR is demonstrated by Western blotting using wild type and knockout embryonic hearts. Loading was monitored by a monoclonal antibody to clathrin. The specificity of Rb80 is also shown in the cryostat section of embryonic hearts in (**A**). Molecular mass standards are indicated on the left of each panel. (**E**) Whole mounts of E10.5 hearts stained by the fiber knob Ad2-Cy2 and Rb80 to illustrate the localization of the CAR. Scale bars (from left to right) 190, 60, and 70 μm. (**F**) Spontaneous beating frequency of E10.5 wild type and CAR-deficient heart organ cultures over a period of 72 h; 10 knockouts and 17 wild type/heterozygote hearts were analyzed (*p* < 0.0001; two-way ANOVA). (**G**) Spontaneous beating frequency of embryonic wild type heart organ cultures over a period of 72 h in the absence or presence of 0.5 mg/mL of fiber knob Ad2. 27 wild type hearts treated with Ad2 and 27 controls were evaluated (*p* < 0.0001; two-way ANOVA). (**H**) Spontaneous beating frequency of embryonic wild type heart organ cultures over a period of 72 h in the absence or presence of 0.5 mg/mL of the IgG fraction of Rb80 to mCAR; 18 wild type hearts treated with Rb80 and 18 hearts as controls were evaluated (*p* < 0.0001; two-way ANOVA).

## Data Availability

Datasets may be found at the Gene Expression Omnibus (GEO) with accession number GSE138831.

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
