# Peer review of "The IgCAM CAR Regulates Gap Junction-Mediated Coupling on Embryonic Cardiomyocytes and Affects Their Beating Frequency"

_life, 2022, doi:10.3390/life13010014_

Round 1

Reviewer 1 Report

1)             At first glance it is counter intuitive that the absence of CAR leads to the decrease expression of Cx43 and Cx45 expression, yet increases the size of connexin clusters in the cell membrane. The authors may want to comment on this.

2) How does the present study compare to those in adult heart where CAR gene deficiency led to reduced expression of Cx43 and Cx45 and disrupted electrical conduction between cardiomyocytes.

Author Response

Please see PDF file

Reviewer 2 Report

The paper entitled “The IgCAM CAR regulates gap junction mediated coupling on embryonic cardiomyocytes and affects their beating frequency” investigates the mechanistic aspects of CAR involvement in regulating the murine heartbeat.

The authors stated in the paper that “CAR ablation has been shown to disrupt electrical conduction between cardiomyocytes in the adult heart and was correlated with reduced expression of Cx45 and Cx43 [17,51]”. My major concern is whether there are any new findings in this study compared to previous ones. The previous studies have also looked into mechanistic aspects of CAR involvement in cardiac function. And they found CAR relation with gap junction formation. And when gap junction is the target point, why studying electrophysiology and Ca cycling in each individual cardiomyocyte in this study! In my point of view, the study should have focused on cell-cell coupling experiments rather than individual cell experiments. However, I should also mention very well designed experiments and wide range of techniques which have been used in this study. And if the question was more related to what they were looking for, it would have been very well conducted study.

My other points are as follow:

Methods:

1.      Where does mouse line CAR (B6.Cg-Cxadrtm1/Fgr) come from. Were they made in the lab or purchased?

2.      In section 2.12, the sentence “cryostat sections were incubated in 0.1% Triton X-100/PBS/1% heat-inactivated goat serum using antibodies listed in supplemental table S3” sounds not right!

Results:

1.      In section 3.1, the authors stated that “These results suggest that the increased beating rate of cultured CAR-deficient embryonic cardiomyocytes could be linked to alterations in calcium cycling, intracellular calcium levels, calcium extrusion from the cytosol, calcium release from internal stores or ionic currents. Alternatively, the increased frequency of calcium transients could result from impaired cell-cell communication. To distinguish between these possibilities, we examined calcium cycling, as well as the electrophysiological properties and cell-cell coupling of WT and CAR-deficient cardiomyocytes. What was the reason behind their hypothesis for impaired Ca cycling by CAR KO! How the Ca machinery could be affected by CAR! I think the answer to this question was clear and that is why they get results which they show in section 3.1.1.

2.      In section 3.1.1. how did the authors measure systolic and diastolic Ca concentrations?

3.      In figure 2C, they checked SR Ca leak. Why CAR should influence SR Ca leak?

4.      In figure 2E, they see changes in the contribution of three Ca extrusion systems in the cardiomyocytes’ Ca cycling (i.e NCX, PMCA and mitochondria). How do they explain this?

5.      In the end of section 3.1.1, the authors stated that “The increased activity in calcium signaling that we measured in CAR KO cardiomyocytes may instead be a reflection of the increased beating frequency in these cells”. In my opinion, the increased Ca cycling is majorly caused by increased electrical activity or beating frequency. I would like to ask the authors to explain whether they have other reasons for this. If yes, explain them.

6.      In figure 3, what is the criteria for a Cx43 cluster? Please define it.

7.      What is the unit in X axis of figure 3C?

8.      In the following sentence, it should be figure 4C and D. “The mRNA levels of Cx43 and Cx45, as well as a number of other genes, including cell-cell adhesion components and cytoskeletal elements, remained unchanged (Figure 3C and D)”.

9.      Why in figure 5H, the beating frequency is still higher with anti-mCAR at each time point?

Discussion:

1.      With the primary hypothesis that CAR should promote gap junction formation, and given the results obtained in this manuscript, the authors has not discussed this discrepancy well. Why a drop in gap junction density resulted in higher spontaneous beating and Ca cycling. Is it just because of wider spatial localization of those existing gap junctions! This has not been very well described.

2.      In the statement of discussion “The stimulatory effect of Ad2 on the beating of heart organ cultures might therefore result from the disruption of CAR-CAR interactions between neighboring cells”. How disruption could strengthen the localization and activity of gap junctions and therefore, results in an increase in beating frequency

Author Response

Please see PDF file

Round 2

Reviewer 1 Report

all points were adequately addressed

Reviewer 2 Report

No comments.